# Physical Layer Security: Channel Sounding Results for the Multi-Antenna Wiretap Channel

**DOI:** 10.3390/e25101397

**Published:** 2023-09-29

**Authors:** Daniel Harman, Karl Knapp, Tyler Sweat, Philip Lundrigan, Michael Rice, Willie Harrison

**Affiliations:** Department of Electrical and Computer Engineering, Brigham Young University, Provo, UT 84602, USA; knapp5@byu.edu (K.K.); tyler.sweat@byu.edu (T.S.); lundrigan@byu.edu (P.L.); mdr@byu.edu (M.R.)

**Keywords:** physical-layer security, MIMO wiretap channel, multiple antennas, channel sounding, secrecy capacity, channel coding

## Abstract

Many physical-layer security works in the literature rely on purely theoretical work or simulated results to establish the value of physical-layer security in securing communications. We consider the secrecy capacity of a wireless Gaussian wiretap channel using channel sounding measurements to analyze the potential for secure communication in a real-world scenario. A multi-input, multi-output, multi-eavesdropper (MIMOME) system is deployed using orthogonal frequency division multiplexing (OFDM) over an 802.11n wireless network. Channel state information (CSI) measurements were taken in an indoor environment to analyze time-varying scenarios and spatial variations. It is shown that secrecy capacity is highly affected by environmental changes, such as foot traffic, network congestion, and propagation characteristics of the physical environment. We also present a numerical method for calculating MIMOME secrecy capacity in general and comment on the use of OFDM with regard to calculating secrecy capacity.

## 1. Introduction

Over the last several decades, advancements in multiple antenna technology and multi-carrier waveforms have resulted in reliable wireless communications with increased bit rates. As advancements continue, the need for increased security has never been more important. Cryptography is the widely accepted method for secure transmission, relying on mathematical and computational complexity. As computation technology increases, such as with quantum computing, and subsequently time required to attack such systems decreases, additional security measures may need to be added [1]. In other situations, such as for Internet of Things (IoT) devices, high-level cryptography may be entirely unreasonable due to limited computational ability or power constraints.

Physical-layer security can offer additional security to that provided by cryptography with strong information-theoretic guarantees [2,3,4]. The additional security helps with situations such as key sharing while simultaneously increasing the eavesdropper’s confusion about the message [5]. Physical-layer security operates through novel uses of signaling [6] and coding techniques [7].

Shannon introduced the idea of *perfect secrecy* [8], which is expounded upon within Wyner’s wiretap model [9]. More recently, work with the multi-input multi-output multi-eavesdropper (MIMOME) Gaussian wiretap channel has been conducted by Khisti and Wornell [10], Oggier and Hassibi [11], and Liu and Shamai [12], yielding a model for the general case of multi-antenna secrecy capacity setups. Despite this information-theoretic foundation, very little work has been conducted in channel sounding to find realistic estimates for the MIMOME secrecy capacity in real-world environments [13]. Work in this area has been conducted with regards to single-input single-output (SISO) systems [14], and SISO vehicle-to-vehicle (V2V) systems [15].

We consider a three-node MIMOME system with two transmit antennas, two receiver antennas, and two eavesdropper antennas, deployed in a highly trafficked indoor environment. We assume a genie-aided system in which the channel state information (CSI) is known at all terminals, and no cryptography is employed. Over a period of time, we capture CSI using off-the-shelf hardware and calculate the secrecy capacity of the environment subject to both temporal and spatial variations to analyze the potential efficacy of physical-layer security in a real-world scenario. We discuss the impacts of the specific environment, such as foot and network traffic, and building construction. We show that these impacts can have significant effects on secrecy capacity over a wide range of SNRs. We also discuss the use of orthogonal frequency division multiplexing (OFDM) in the 802.11n WiFi standard with regard to calculating secrecy capacity. Finally, we provide a numerical method to optimize a brute force search to calculate MIMOME secrecy capacity in general, for any arbitrary SNR. Note that this work does not include a discussion or prescription for deploying channel coding in the case of the MIMOME channel, as no conclusive work yet exists as to what class of codes to use, and such a work would merit a number of publications of its own.

While many of the works listed previously rely on purely theoretical approaches to physical-layer security, this work analyzes actual channel sounding results captured in a real environment. Coupling the theoretical foundation with real experimental measurements, we demonstrate the potential efficacy of a MIMO physical-layer scheme in an indoor office-like environment using existing commercial hardware employing WiFi, an existing protocol. This is a vital next step in realizing the widespread use of physical-layer security, as it will hopefully inform future research and development efforts about the expected performance of any complete physical-layer security system. Additionally, there is little work on calculating MIMOME secrecy capacity in general, and this paper provides a simple, low-complexity algorithm to accomplish this.

The paper is organized as follows. Section 2 is a brief explanation of the notion used. Section 3 presents the wiretap channel model and assumptions used to calculate the secrecy capacity, including the details of our optimization algorithm for calculating MIMOME secrecy capacity. The procedure for acquiring the CSI, the physical environment and other variables are discussed in Section 4. We make concluding remarks in Section 5.

## 2. Notation

Bold font lower-case characters are reserved for vectors. Bold font capital letters represent either matrices or random variables, made clear by the context. Vector and matrix dimensions are either denoted as superscripts or made clear within the text. Vector dimensions also explicitly state whether each vector is a row or column vector by use of 1×n or n×1, respectively. Subscripts usually connect a variable to a user within the wiretap model, with a few exceptions. For example, $C_{s}$ denotes secrecy capacity and $k_{i}$ denotes an element, of index *i*, within a vector. I denotes the identity matrix and 0 denotes the zero matrix, where the respective dimensions are made clear by context. We let $C^{m\times n}$ represent the set of all m×n dimensional complex matrices. Circularly symmetric complex Gaussian random variables with mean zero and covariance matrix I are denoted as $\mathrm{CN}\left(0,I\right)$. Also, we use range(x,y,z) to denote a vector starting at *x*, ending at *y* and of length *z*. Where *z* is not denoted, it is left for the context to determine. Furthermore, all logarithms are assumed to be base 2.

Additionally, we use $\det\left(\cdot \right)$ and $\mathrm{tr}\left(\cdot \right)$ to denote the determinant and trace of a matrix while $\mathrm{diag}\left(\cdot \right)$ denotes a diagonal matrix whose argument determines the diagonal elements of the matrix. The superscripts ${}^{T}$ and ${}^{†}$ are used as the matrix transpose and the conjugate transpose operators, respectively. ||·|| denotes the Euclidean vector norm, and $\left|\right|\cdot {\left|\right|}_{F}$ denotes the Frobenius norm of a matrix. Finally, we use A⪰0 to denote a positive semi-definite matrix A.

## 3. Channel Model and System Setup

The framework for our experimental model is based on Wyner’s 1975 work on the wiretap channel [9]. The model assumes three users: Alice, the legitimate transmitter; Bob, the legitimate receiver; and Eve, a passive eavesdropper. Alice transmits an encoded message as x to Bob over the main channel. Bob and Eve receive $y_{m}$ and $y_{e}$, respectively, which are noisy versions of the input x; see Figure 1. We assume that the main and eavesdropper’s channels are time-varying, but are static for the duration of an individual transmission. Additionally, our setup features the use of multiple antennas for all nodes. As for the wiretap setup, we assume that Eve is capable of receiving and decoding messages sent from Alice. Finally, both Bob and Eve have an OFDM demodulator with cyclic prefix removal and FFT [16,17].

### 3.1. MIMOME Secrecy Capacity

The work in [10,11] independently developed the model for the MIMOME wiretap channel. In this work, we let the channel models for Bob and Eve be represented by
(1)$$\begin{array}{cc} y_{m}\left(t\right) & =H_{m}\left(t\right)x\left(t\right)+z_{m}\left(t\right)\\ y_{e}\left(t\right) & =H_{e}\left(t\right)x\left(t\right)+z_{e}\left(t\right) \end{array}$$
respectively, where $x\left(t\right)\in C^{n_{t}\times 1}$ is the OFDM transmitted signal vector at time *t*, and $H_{m}\left(t\right)\in C^{n_{m}\times n_{t}}$ and $H_{e}\left(t\right)\in C^{n_{e}\times n_{t}}$ are the channel gain matrices of the main and eavesdropper channels, respectively, at time *t*. We use $n_{t}$, $n_{m}$, and $n_{e}$ to denote the number of transmitter, main receiver and eavesdropper antennas. The noise, at time *t*, is represented by $z_{m}\left(t\right)\in C^{n_{t}\times 1}$ and $z_{e}\left(t\right)\in C^{n_{t}\times 1}$, which are independent and identically distributed (i.i.d.) as CN(0,I).

The channel input is subject to the power constraint
(2)$$E\left\{{\left||x\left(t\right)|\right|}^{2}\right\}\leq P$$
where *P* is the total power constraint and the expectation is taken with regard to time. For brevity, the remainder of this work will omit function notation with regard to time, unless otherwise noted. The secrecy capacity is defined as the supremum of all achievable rates that satisfies the power constraint and the min-max optimization problem, i.e., maximizing the main channel capacity while minimizing the eavesdropper channel capacity. Secrecy capacity is expressed as
(3)$$C_{s}=\max_{K_{x}}\left(\log\frac{\det\left(I+\frac{1}{\sigma_{m}^{2}}H_{m}K_{x}H_{m}^{†}\right)}{\det\left(I+\frac{1}{\sigma_{e\;\;}^{2}}H_{e\;\;}K_{x}H_{e\;\;}^{†}\right)}\right),$$
where the main and eavesdropper channels have noise variances $\sigma_{m}^{2}$ and $\sigma_{e}^{2}$, respectively. We use $K_{x}$ to denote the covariance matrix of x, i.e., $K_{x}=E\left\{xx^{†}\right\}$. The power constraint (Equation 2) can be written with $K_{x}$, such that
(4)$$K_{x}≜\left\{K_{x}:K_{x}⪰0,\;\mathrm{tr}\left(K_{x}\right)\leq P\right\}\;.$$

Note that all covariance matrices are positive-semi-definite by definition; however, we explicitly state this property for clarity. As noted in [18], the total power channel gain is defined as ${\left||H|\right|}_{F}$, where H is a channel gain matrix. This does, however, only apply in conditions where the total transmit power is equal for all transmit antennas. As the transmitter in general does not have equal power across all transmit antennas, and to maintain a consistent metric for both the main and eavesdropper channels, we define SNR for the MIMO case [19] to be
(5)$$\mathrm{SNR}=\frac{P}{\sigma_{n}^{2}}$$
where, $\sigma_{n}^{2}=\sigma_{m}^{2}=\sigma_{e}^{2}$.

### 3.2. High-SNR Asymptotic Results

We exploit the high-SNR asymptote from [10] in our analysis to check the accuracy of the brute force search described below. In the high-SNR regime, the secrecy capacity approaches a non-zero threshold for the maximum achievable secure rate. The generalized singular value decomposition (GSVD) is used to decompose the channel gain matrices into an equivalent parallel channel model. We define $m_{i}$ and $e_{i}$ to be the individual generalized singular values of the main and eavesdropper channels, respectively. We follow the convention of choosing the arbitrary indexing where singular values are ordered by ascending magnitude. Thus, we assign the generalized singular values as,
(6)$$\sigma_{i}≜\frac{m_{i}}{e_{i}},\;i=1,2,\cdots ,s,$$
where *s* is the total number of singular value pairs. The high-SNR regime asymptote for secrecy capacity is then given as
(7)$$\lim_{P\to \infty }C_{s}\left(P\right)=\sum_{j:\sigma_{j}\geq 1}\log\sigma_{j}^{2},$$
because of the condition given in [10] that $\mathrm{rank}\left(H_{e}\right)=n_{t}$.

### 3.3. Note on Multi-Carrier Waveforms

A hallmark of the IEEE 802.11 standards for WiFi has been its use of OFDM, increasing both rates and reliability. Our approach to the MIMOME secrecy capacity includes the use of all 56 subcarriers available in the 802.11n standard. There has been some work conducted on the OFDM water-filling problem [20] for the MIMOME secrecy capacity with multi-carrier waveforms with a particular focus on energy efficiency; however, more work still remains to be conducted for an efficient solution that maximizes secrecy rate. Solutions designed around maximizing secrecy rate would begin with a total power constraint and then optimally allocate power to each of the main channel’s subcarriers’ eigenmodes while simultaneously minimizing the power transmitted to the eavesdropper, [16,21,22]. With a protocol such as any of the 802.11 standards and enough antennas, this can prove to be a daunting optimization problem, even when employing efficient iterative numerical methods designed for the non-convex, non-smooth nature of the optimization. If such solutions are to be deployed in real-world systems, it would be expected from previous work that the secrecy capacity would be significantly higher.

Thus, due to this constraint, we consider each subcarrier channel as having its own power constraint equal to the power constraints of the other subcarriers. The total secrecy capacity is then considered to be the sum of the secrecy capacity of all available subcarriers, defined below as,
(8)$$C_{s}^{\mathrm{OFDM}}=\sum_{i=1}^{q}C_{s}\left(i\right)$$
where *i* is the index of subcarrier, and *q* is the total number of subcarriers.

### 3.4. Brute-Force Optimization

This section outlines our technique for calculating MIMOME secrecy capacity based on a brute force algorithm. There exist other techniques in the literature that either focus on minimizing power [23] or gradient descent algorithms for specific cases [24]. The algorithm presented here is simple and based on constraining the search region under the assumption that the transmitter has a fixed power input as in (Equation 4). The algorithm presented here could be paired with the algorithm presented in [24] to better determine the starting criteria for the gradient descent method. This paper’s novel contribution to this area lies mainly in its ability to constrain the search region using the method described below, which is an essential step whatever the final minimization technique.

As noted in [25], the objective function in the secrecy capacity expression (Equation 3) in general does not satisfy the Karush–Kuhn–Tucker (KKT) conditions; thus, a brute force search is used to identify the optimal input covariance matrix, $K_{x}$. During our analysis, we found that the most computationally expensive cases can be predetermined by the GSVD analysis proposed in [10] and eigenvalue analysis proposed in [11]. The work conducted in [11] considers two cases, which they refer to as the *definite* and *indefinite* cases. The *definite case* is defined by the following, which corresponds to a definite advantage for either the main or eavesdropper channels, respectively:(9)$$H_{m}^{†}H_{m}≻H_{e}^{†}H_{e},$$
(10)$$H_{e}^{†}H_{e}≻H_{m}^{†}H_{m}.$$ The definite case corresponds to the condition in (Equation 7), where all generalized singular values are greater than one. The *indefinite case* is where a clear advantage is not present for either the main or eavesdropper’s channel. It is defined where both of the following expressions, (Equation 11) and (Equation 12), are true, i.e., $H_{e}^{†}H_{e}-H_{m}^{†}H_{m}$ has positive eigenvalues in addition to others that are zero or negative. Precisely,
(11)He†He≽Hm†Hm,
(12)$$H_{m}^{†}H_{m}⊁H_{e}^{†}H_{e}.$$ The indefinite case corresponds to the condition in (Equation 7), where only some of the generalized singular values are greater than one.

The indefinite case is where we found the most difficulty with the brute force search. We found this analysis to be a natural starting point to determine the needed precision in the optimization, as checking for either the definite or indefinite case is computationally inexpensive relative to a high-precision search. In the definite case, lower precision is often sufficient to optimize to the desired level of accuracy. In the indefinite case, a substantially higher level of precision is required, as the gradient often has much more extreme changes around local and global extrema.

To simplify the brute force search, we restrict the search area to a region that satisfies all problem constraints; see (Equation 4). We assume that the optimal solution involves using all available power, which may not hold for all scenarios but it held true for our measurements and setup. This assumption makes the expression (Equation 4) solely an equality, restated below in (Equation 14). Given that our experimental data was gathered using NICs where $n_{t}=n_{m}=n_{e}=2$, we define the following degrees of freedom in the search, such that
(13)$$K≜\left[k_{1},k_{2},k_{3}\right],$$
where
(14)$$K_{x}≜\left\{K_{x}:K_{x}⪰0,\;\mathrm{tr}\left(K_{x}\right)=P\right\}$$
and
(15)$$0\leq k_{1}\leq P,$$
such that
(16)$$K_{x}=\left[\begin{array}{cc} k_{1} & k_{2}-jk_{3}\\ k_{2}+jk_{3} & P-k_{1} \end{array}\right].$$

We begin by creating a set of nested for loops, iterating over each combination of all degrees of freedom with a hierarchy described by the order in (Equation 13). A closed-form expression can be found that restricts the search area for each iteration of $k_{1}$ by exploiting the quadratic formula during the process of finding the eigenvalues of $K_{x}$, ensuring the positive-semi-definite condition in (Equation 14).

We begin with the identity of the eigendecomposition,
(17)$$|K_{x}-\lambda I|=0.$$
We let,
(18)$$K_{\lambda }=\left[\begin{array}{cc} k_{1}-\lambda & k_{2}-jk_{3}\\ k_{2}+jk_{3} & P-k_{1}-\lambda \end{array}\right]$$
Combining terms and simplifying yields
(19)$$|K_{\lambda }|=0.$$
Next, we solve for the characteristic polynomial, i.e.,
$$\det K_{\lambda }=\lambda^{2}-\lambda P+k_{1}P-k_{1}^{2}-k_{2}^{2}-k_{3}^{3}$$
and utilize the quadratic formula to solve for the roots of the characteristic polynomial, given as
(20)$$\lambda =\frac{P\pm \sqrt{P^{2}-4\left(k_{1}P-\left(k_{1}^{2}+k_{2}^{2}+k_{3}^{3}\right)\right)}}{2}.$$
Analysis of (Equation 20) reveals that λ≥0 if
(21)$$\;k_{1}P\geq k_{1}^{2}+k_{2}^{2}+k_{3}^{2},$$
thus ensuring a positive semi-definite condition.

With the search area constrained, it is advantageous to search the edge cases, as we found that the edges often contain local extrema. From the above expression, the maximum radius, $\tilde{r}$, is calculated as
(22)$$\tilde{r}=\sqrt{k_{1}\left(P-k_{1}\right)}.$$ A radial or grid search is then employed, where $k_{2}$ and $k_{3}$ are constrained by (Equation 21) and (Equation 22). A grid search produces $k_{2}$ and $k_{3}$ in $\mathrm{range}(-\tilde{r},\tilde{r})$. A radial search produces *r* and θ in $\mathrm{range}(0,\tilde{r})$ and range(0,2π), respectively. We then convert *r* and θ from polar to rectangular coordinate systems, producing the appropriate values of $k_{2}$ and $k_{3}$. Iterating $k_{1}$ in range(0,P) over a sufficiently large interval provides the required level of accuracy. The accuracy of the search can be checked against the high-SNR asymptotic approach (Equation 7).

This closed-form expression is usually computationally cheaper than the ill-conditioned problem of calculating eigenvalues to ensure the positive-semi-definite condition (Equation 14). In our experience, using this method and the radial search described previously was approximately 3600 times faster than an ill-targeted brute force search based on timers in our code. This method can be exploited for higher dimensional covariance matrices; however, it is limited by the closed-form solution to solving for the roots of a polynomial expression.

## 4. Channel Sounding

### 4.1. Environments

A temporal wiretap scenario simulating a passive eavesdropper attack was implemented in the step-down lounge study area of the Clyde Building at Brigham Young University (see Figure 2 for building layout). The area where data were gathered is open and the walls are cinder block with many windows on the west-facing wall. On the north end, there is an area partitioned by a glass wall in which Alice and Bob were set up about 10 m apart. Eve was set up outside the study area approximately 33.5 m from Alice to sufficiently disadvantage the eavesdropper while maintaining line of sight.

A set of space-varying experiments were implemented in two parts on both the fourth floor of the Clyde Building and the fourth floor of the Engineering Building. The experimental area in the Clyde Building is a hallway with windows looking into student lab areas on either side. The hallway’s primary material composition is that of reinforced glass windows, cinder block walls, and vinyl flooring tiles. The experimental area in the Engineering Building took place in the west electrical engineering faculty hallway. It consists of modern building materials, with metal framing, sheetrock covering, and carpeted flooring. Measurements were taken in a grid pattern with 12 cm spacing.

Measurements for the temporal experiment were taken between the hours of 11:30 a.m. and 3:30 p.m., during which time there were students utilizing the study space. Measurements for the spatial experiment were taken between the hours of 9:00 a.m. and 5:00 p.m. over the course of several weeks. Additionally, because the channel sounding measurements were taken in an indoor environment, we assume that the thermal noise for both the main and eavesdropper channels was equal.

### 4.2. Materials and Methods

The main goal of this paper was to use off-the-shelf radio equipment where possible to ensure realistic results. Thus, each node in the experiment runs Ubuntu 14.04 with a custom kernel developed by Xie [26] and network interface cards (NICs) based on the Qualcomm Atheros AR938X chipset. This design decision is based on using an existing, tested setup for measuring CSI on commercial off-the-shelf hardware to ensure results that are as realistic as possible. The standard 802.11n was employed due to the NICs and custom kernel support; however, extending our results to other WiFi or wireless standards was tied directly to the access of the internal channel sounding measurements taken by the hardware. Each node (a desktop computer running the modified Ubuntu kernel with the Atheros NIC) was deployed on a mobile cart for easy repositioning. The orientation of the cart was kept the same for all the spatial experiments, preventing measurements immediately adjacent to the walls as evidenced in the 1st and 2nd figures in Section 4.3.2.

Alice is configured as a WiFi access point (AP) and sends out a user datagram protocol (UDP) packet every 30 s for the temporal experiment and every half second for the spatial experiment. For the spatial experiment, every location’s secrecy capacity result is a time average of the individual secrecy capacity from 150 CSI measurements. Both Bob and Eve are configured as clients to the AP and upon receiving a packet, and the kernel calculates and records the CSI for each antenna pair and every OFDM subcarrier.

Following data acquisition, a brute force optimization was run for each packet to calculate the secrecy capacity across all subcarriers as outlined in Section 3.4. Due to practical issues with off-the-shelf hardware and the environment, not all packets were received at either receive node, thus not triggering a CSI measurement in those situations. To account for this, we assume 0 to be the complex channel gain matrix for either Bob or Eve in these situations. This means that for situations when the eavesdropper fails to capture the packet, the secrecy capacity is considered to be simply the capacity of the main channel. In situations where the main channel fails to capture a packet, the secrecy capacity is simply zero because there is no main channel achievable rate to begin with.

### 4.3. Channel Sounding Results and Discussion

#### 4.3.1. Temporal Experiment

The results of our temporal experiments are shown in Figure 3 for a single instance of a subcarrier, and for the whole experiment in Figure 4. The results in Figure 3 show a basic example where both receiver nodes have non-zero CSI data. The high-SNR asymptote is calculated using (Equation 7) to demonstrate the accuracy of the optimization.

As can be seen, at high SNR it is possible to achieve significant bit rates comparable to normal usage. At very low SNR, it is much more difficult to get meaningful security. We note here that we assume the noise variance is unity and equal for the main and eavesdropper channels. We have chosen a unity noise variance to stay consistent with [10,11] and for numerical stability; however, as can be seen by inspection of (Equation 7), the noise power does not contribute to the secrecy capacity in the high-SNR regime.

The results in Figure 4 denote the total OFDM MIMOME secrecy capacity as a function of both time and SNR. An interesting observation is that during the first 3800 s of the capture, the secrecy capacity in the high-SNR regime is much lower (approximately an average of 500 bits per channel use) than afterward (approximately an average of 1500 bits per channel use). In the beginning, the reduced secrecy capacity was likely caused by random environmental factors, e.g., a student was studying with a laptop connected to another WiFi AP, which would have influenced the main channel capacity. After the student left, the interference to the main channel presumably diminished. Thus, issues of security are not always a matter of disadvantaging the eavesdropper as they are increasing the advantage of the main channel. During the latter portion of the experiment, Eve was unable to receive approximately a fourth of the packets transmitted. This resulted in the zero-filled CSI condition mentioned previously. Thus, for the portions of the graph with the highest secrecy capacity, the values are equivalent to the Shannon capacity calculation for Bob. The graph shows high achievable secrecy rates for the majority of the experiment. These results suggest that environmental effects greatly influence security, especially in the MIMOME setup, for better or for worse.

#### 4.3.2. Spatial Experiment

The results for the Clyde Building secrecy capacity experiment are shown in Figure 5 and the results for the Engineering Building are shown in Figure 6. On each figure, Bob and Alice are fixed and each data point measures the secrecy capacity for the location as a potential position for Eve. These two experimental environments were chosen primarily for their differences. They both feature an unsecured hallway, which poses a large problem for any security scenario employing a wireless network due to RF signal propagation through walls and structures.

From our analysis of Figure 5 and Figure 6, we note in instances of direct line-of-sight between the eavesdropper and the transmitter there are often substantial drops in the resulting secrecy capacity, a difference of approximately 500 bits per channel use in the most extreme cases. We also note that secrecy capacity tends to increase in proportion to the distance of Eve from the transmitter. The figures show that in instances of near equivalent channels between Bob and Eve, meaningful secrecy capacity can still be achieved (of approximately 100 to 200 bits per channel used in each building where the distances from Alice are roughly equal for Bob and Eve), which we note as a fundamental result of the measurement campaign.

Physical obstacles also play a fundamental role in the indoor scenario as they contribute to non-uniform or rapidly changing channel effects. Examples from the figures include office furniture, walls, doors, pillars, etc. These effects are both to the advantage and the detriment of the intended receiver in the measured environments, but could be placed strategically to increase secrecy capacity. As stated before, these two environments differ in the types of propagation characteristics that they exhibit due to their construction. The Clyde Building was opened in 1974 and the Engineering Building was opened in 2018. They have different material compositions and were built by employing different construction methods. RF propagation in the WiFi bands is easier in buildings that use sheetrock walling rather than the more durable cinder block walls. Our analysis showed that building construction and materials are important security considerations, as can be noted by comparing the unsecured hallways between the two environments. Both instances have cases where Eve has a presumably superior channel; however, the secrecy capacity is generally higher for those cases in the Engineering Building, likely due to the cinder block construction.

Physical-layer security, and security in general, requires a specific understanding of the scenario, environment, network users and threat models. The variety of ways that malicious activity can be carried out requires security experts to be more conscious of the specifics of attacks. The spatial experiments carried out are an example of the utility that a measurement campaign can have in determining weaknesses.

## 5. Conclusions

In this paper, CSI was recorded over time and space in several indoor environments using real-world, off-the-shelf WiFi NICs. An experiment was conducted to determine the secrecy capacity for a range of SNR and plotted as a temporal heatmap. Two additional experiments were conducted to determine the secrecy capacity as a function of space, and plotted as a spatial heatmap against the layout of the environments in which the experiments took place. Our results show that environmental changes, such as foot and network traffic, physical obstacles, and relative positions to the transmitter affect the maximum achievable secure rate of communication significantly (an average of 1000 bits per channel use difference between the interference and non-interfered cases of the temporal experiment and by as many as 500 bits per channel use in the spatial experiments due to physical obstacles or relative positions to Alice). It is shown that meaningful and secure rates can theoretically be achieved in real and practical situations using commercially available hardware, as evidenced by the non-zero secrecy capacity in even non-ideal situations.

## Figures and Tables

**Figure 1 entropy-25-01397-f001:**
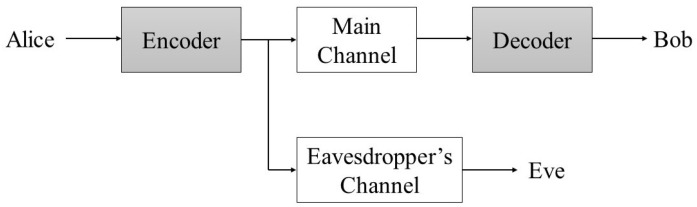
Wyner’s wiretap model.

**Figure 2 entropy-25-01397-f002:**
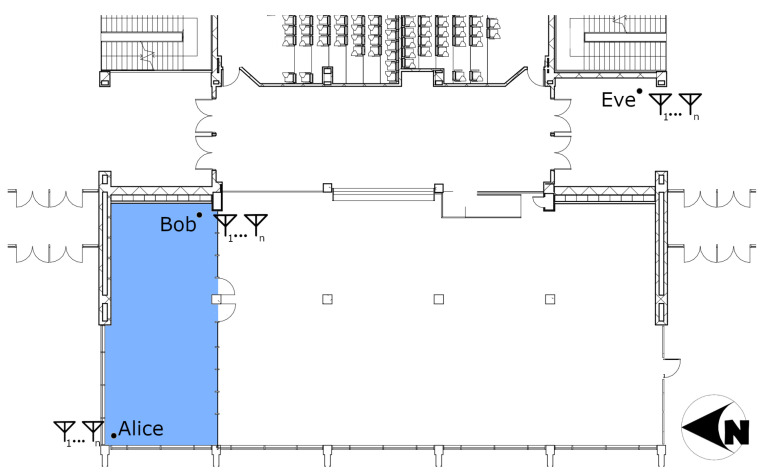
Indoor environment where the experiment was staged. The blue area is partitioned off from the main area by a glass wall and door.

**Figure 3 entropy-25-01397-f003:**
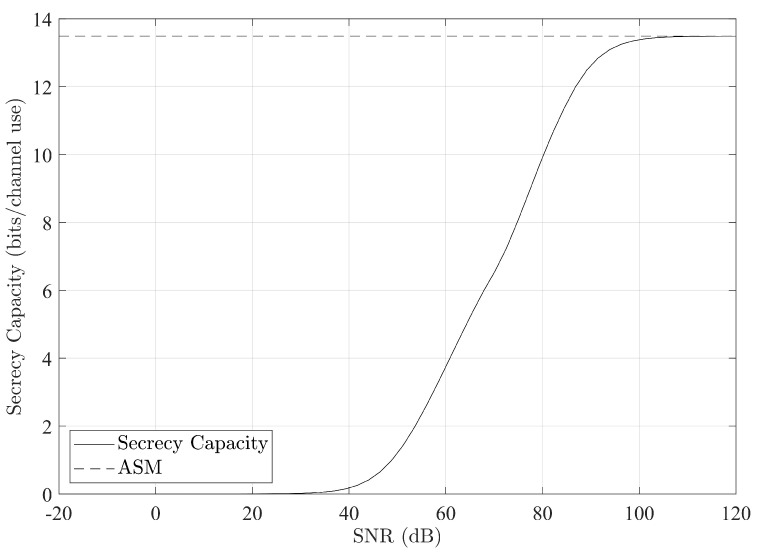
Secrecy capacity for a single subcarrier’s set of measurements calculated over a range of transmit powers. ASM is the high-SNR asymptote for secrecy capacity as described in (Equation 7).

**Figure 4 entropy-25-01397-f004:**
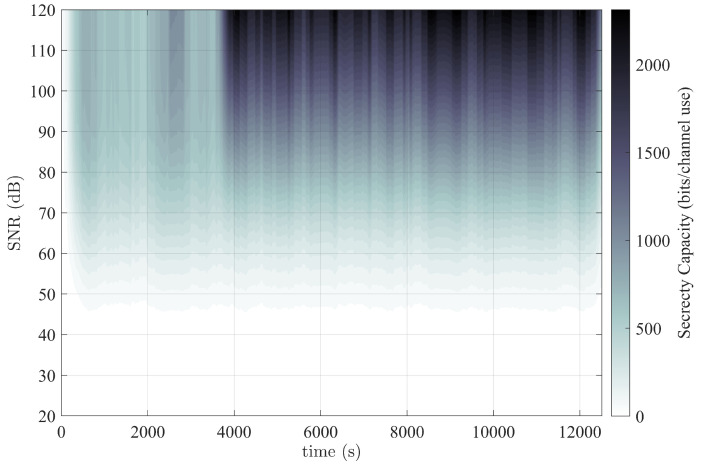
Experimental results over a period of 3.5 h as a function of time and SNR denoting the secrecy capacity in bits/channel use.

**Figure 5 entropy-25-01397-f005:**
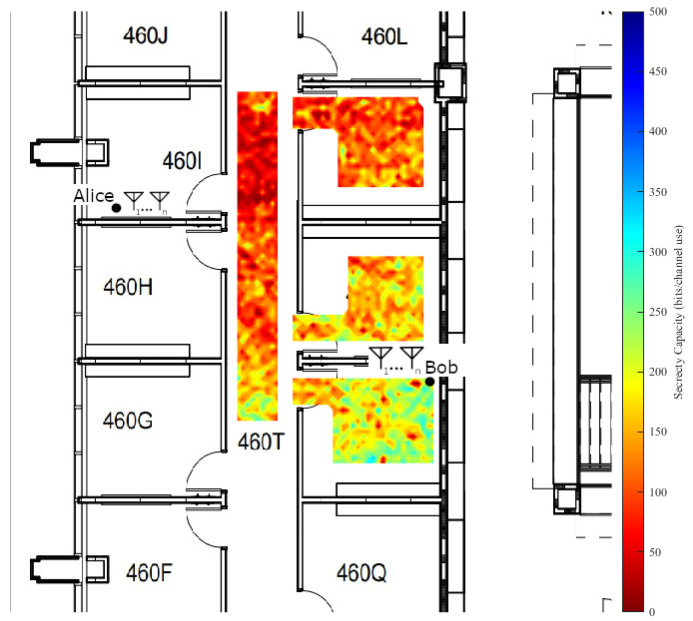
Spatial heatmap of average secrecy capacity in the Engineering Building.

**Figure 6 entropy-25-01397-f006:**
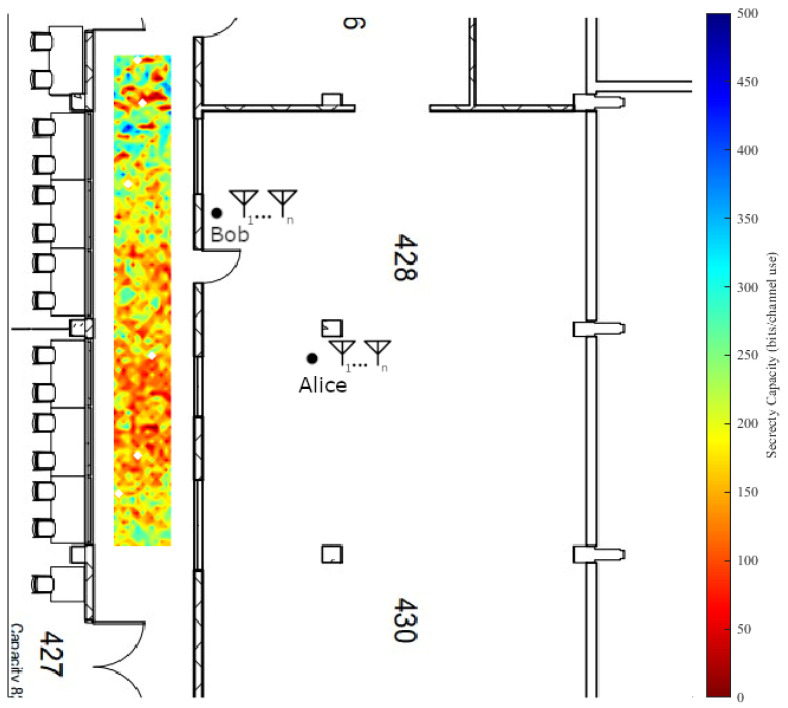
Spatial heatmap of average secrecy capacity in the Clyde Building.

## Data Availability

Channel sounding measurements and related code available upon email inquiry to the authors.

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
