# Peer review of "Physical Layer Security: Channel Sounding Results for the Multi-Antenna Wiretap Channel"

_entropy, 2023, doi:10.3390/e25101397_

Round 1

Reviewer 1 Report

In this paper, the authors analyze the secrecy capacity of a wireless Gaussian wiretap channel in a real-world scenario, deploying a multi-input, multi-output, multi-eavesdropper (MIMOME) system over an 802.11n wireless network. They highlight the substantial impact of environmental factors like foot traffic, network congestion, and physical propagation characteristics on secrecy capacity, while also introducing a numerical method for its calculation and discussing the use of orthogonal frequency division multiplexing (OFDM) for this purpose.

Comments and Suggestions to the Authors:

1.       The introduction requires a paragraph that succinctly highlights the exceptional aspects of the authors' work and the innovations it brings compared to existing research. At present, the introduction lacks clarity in revealing the scientific novelty of the proposed research. Emphasizing the novelty/originality would better engage the readers and provide a clear direction for the study.

2.       I think the quality and resolution of the pictures should be improved, especially Figure 1, Figure 3, and Figure 4. Also, the formatting of Figure 5 and Figure 6 in the text seems strange.

3.       In the conclusion section, include a quantitative assessment of the improvements achieved by this research compared to the works of other researchers. Presenting concrete data will strengthen the conclusions and provide valuable insights. Because current conclusions are very abstract.

I'd suggest accepting this paper with major revisions after addressing the comments and suggestions above.

Reviewer 2 Report

Author propose a numerical method to optimize a brute-force search to calculate MIMOME secrecy capacity in general, for any arbitrary SNR. 

The method proposed exhibits promising efficiency improvements, making it a significant advancement in the realm of secure wireless communications.

However, while the paper stands as a commendable achievement in its own right, it could further enhance its depth and impact by delving into more extensive comparisons with related works in the literature. Exploring how this optimized approach stacks up against existing methods in various scenarios and under different system parameters could provide valuable insights and a more comprehensive understanding of its strengths and weaknesses.

- Other questions about the manuscript with important answers could be provided.

How can the methodology of the research work on optimizing the numerical method for calculating MIMOME secrecy capacity be improved to enhance its overall quality and effectiveness?

What are the key components of the numerical method proposed by the author to optimize the brute-force search for calculating MIMOME secrecy capacity at various Signal-to-Noise Ratios (SNR), and how does it contribute to a more efficient and general approach?

Is the method proposed by the author to optimize the secrecy capacity calculation in 802.11n OFDM systems easily adaptable and applicable to other versions that employ OFDMA, such as 802.11ax? If so, what are the main challenges and modifications required for its effective implementation in these versions?

Round 2

Reviewer 1 Report

The authors have effectively addressed the majority of my concerns, and I believe the paper is suitable for acceptance in its current state.

Reviewer 2 Report

Dear author, 

Thank you so much for your reply process. 

Also, authors could consider to improve quality presentation of images.